# Gelatinous filter feeders increase ecosystem efficiency
Michael R. Stukel [1,2,6] ✉, Moira Décima [3,6], Christian K. Fender[1], Andres Gutierrez-Rodriguez [4] & Karen E. Selph [5]

Gelatinous filter feeders (e.g., salps, doliolids, and pyrosomes) have high filtration rates and can feed at predator:prey size ratios exceeding 10,000:1, yet are seldom included in ecosystem or climate models. We investigated foodweb and trophic dynamics in the presence and absence of salp blooms using traditional productivity and grazing measurements combined with compound-specific isotopic analysis of amino acids estimation of trophic position during Lagrangian framework experiments in the Southern Ocean. Trophic positions of salps ranging 10–132 mm in size were 2.2 ± 0.3 (mean ± std) compared to 2.6 ± 0.4 for smaller (mostly crustacean) mesozooplankton. The mostly herbivorous salp trophic position was maintained despite biomass dominance of ~10-µm-sized primary producers. We show that potential energy flux to >10-cm organisms increases by approximately an order of magnitude when salps are abundant, even without substantial alteration to primary production. Comparison to a wider dataset from other marine regions shows that alterations to herbivore communities are a better predictor of ecosystem transfer efficiency than primary-producer dynamics. These results suggest that diverse consumer communities and intraguild predation complicate climate change predictions (e.g., trophic amplification) based on linear food chains. These compensatory foodweb dynamics should be included in models that forecast marine ecosystem responses to warming and reduced nutrient supply.

Climate change is predicted to reduce global marine phytoplankton productivity through increased stratification and commensurately reduced nutrient supply[1,2]. Declines in higher trophic levels (e.g., large-bodied copepods) have already been observed and linked to decreasing copepod-mediated carbon sequestration[3]. Climate change, in combination with fishing pressure, is also expected to alter the global marine biomass-size spectrum[4]. However, future impacts of reduced primary production on other taxa and ecosystem services will be mediated by complex food web alterations that are difficult to predict. Modeling studies have suggested a pattern of "trophic amplification", in which declines in higher trophic levels (e.g., fish) are greater than declines in primary production[5–8]. This trophic amplification results from altered plankton size structure and commensurately longer food chains, among other processes[7,9]. The net result is that global biomass of >10-cm marine animals are predicted to decline ~5% °C$^{-1}$ of warming, although biomass declines will be regionally variable[7].

Predictions of future climate change impacts on net primary productivity are grounded in detailed knowledge of the physical, chemical, and physiological drivers of photosynthesis and can be continuously tested using satellite observations[2,10,11]. In contrast, higher trophic level observations are much scarcer, and biomass increases or declines of these taxa are determined by changes in ecosystem efficiency[12–14]. Here, ecosystem efficiency refers to the relative proportion of net primary production (NPP) that is converted into secondary production of top trophic levels, such as commercially valuable fish. Ecosystem efficiency, in turn, depends on trophic efficiency (the ratio of production at one trophic level relative to the trophic level immediately below[15]) and the length of the food chain separating primary producers from top trophic levels. Generally, these food chains are longer in oligotrophic regions dominated by small cyanobacteria and the microbial loop than in productive upwelling domains[13,16]. Greater oligotrophy in a future, more-stratified ocean is thus expected to increase food chain length and decrease ecosystem efficiency.

The impacts of decreased NPP and a shift toward smaller phytoplankton can be offset, however, by foodweb alterations (Fig. 1). Marine planktonic herbivore communities are incredibly diverse, with highly

[1]Department of Earth, Ocean, and Atmospheric Science, Florida State University, Tallahassee, FL, USA. [2]Center for Ocean-Atmospheric Prediction Studies, Florida State University, Tallahassee, FL, USA. [3]Scripps Institution of Oceanography, University of California San Diego, San Diego, CA, USA. [4]National Institute of Water and Atmospheric Research (NIWA), Wellington, New Zealand. [5]Department of Oceanography, University of Hawaii at Manoa, Honolulu, HI, USA. [6]These authors contributed equally: Michael R. Stukel, Moira Décima. ✉e-mail: mstukel@fsu.edu

**Fig. 1 | Conceptual food web diagrams for a size-structured ecosystem or an ecosystem with diverse omnivores and variable predator:prey size ratios.** Conceptual food web diagrams for a size-structured ecosystem with fixed predator:prey size ratios (**a, b**) or diverse omnivores with highly variable pre-dator:prey size ratios (**c, d**) in a large-phytoplankton-dominated system (**a, c**) or a small-phytoplankton-dominated system (**b, d**). The color of circles is proportional to the production (primary or secondary) of a functional group. The trophic amplification hypothesis is based on conventional size-structured ecosystem models (**a, b**). Thus, shifts toward small phytoplankton in a future climate (represented by moving from **a** to **b**) lead to food chain elongation through the insertion of additional protistan trophic levels. In contrast, the compensatory foodweb dynamic suggests that bottom–up processes driving a shift from large (**c**) to small (**d**) phytoplankton would be accompanied by a shift of metazoan communities toward filter feeders with large predator:prey ratios (e.g., salps). This conceptualization of the foodweb involves high functional diversity amongst consumer trophic levels and substantial intraguild predation. These processes could stabilize ecosystem functions in response to climate change disruptions of nutrient supply.

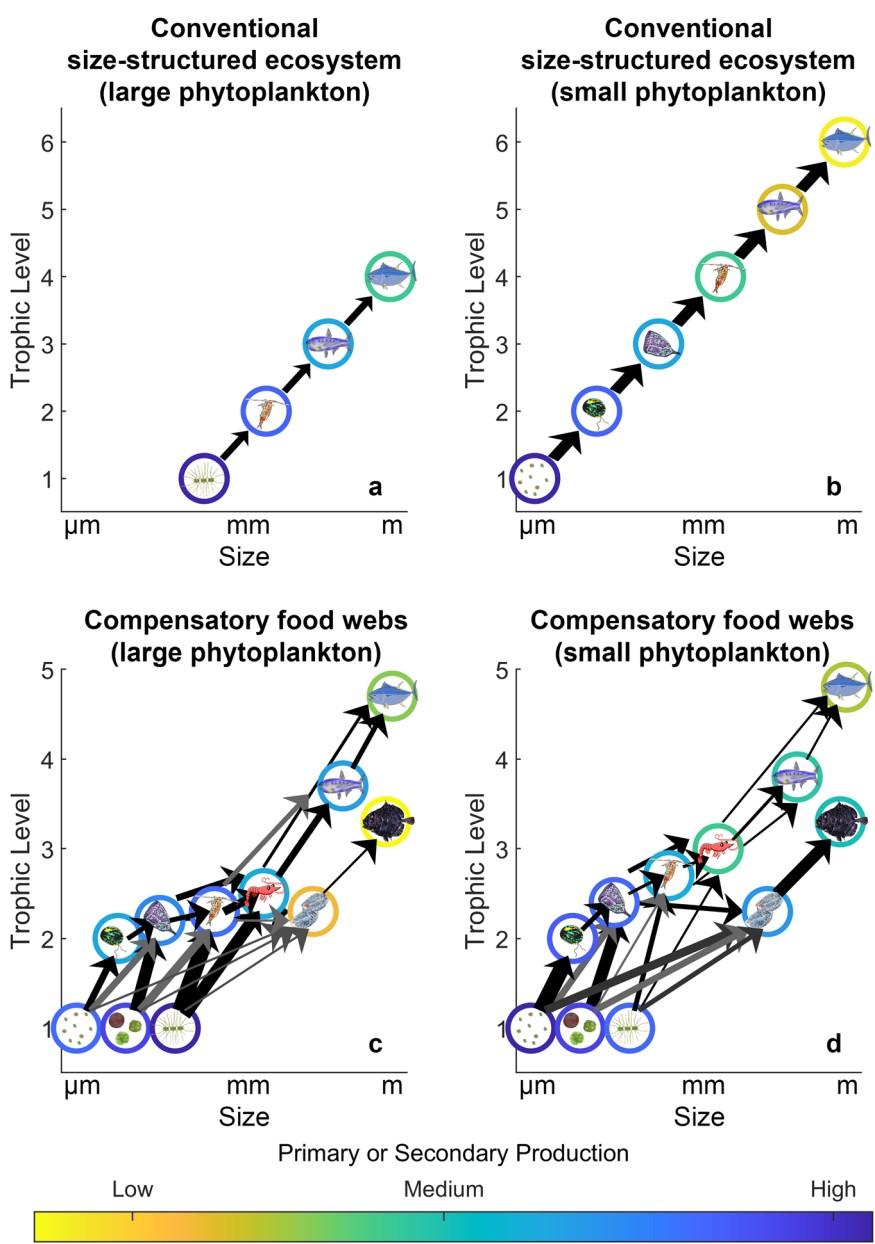

variable predator:prey size ratios; protistan grazers often have predator:prey size ratios varying from 1:1 to 10:1[17–19], omnivorous crustacean often have ~100:1 ratios[17], and pelagic tunicates (salps, doliolids, pyrosomes, and appendicularians) can have values exceeding 10,000:1[20,21]. The relative importance of microbial trophic steps also decreases at higher trophic levels[22]. Taken together, this suggests the possibility for compensatory food-web changes in which a shift to smaller phytoplankton taxa would be accompanied by greater dominance of herbivores with a higher pre-dator:prey size ratio. Such a shift could offset changes predicted by trophic amplification theory and stabilize food webs in response to altered nutrient supply. Indeed, models suggest that gelatinous filter feeders could increase in abundance as a result of climate change thus ameliorating food chain length increases that would otherwise result from a shift toward picophytoplankton[23].

Here, we took advantage of predictable regions of salp dominance near the Chatham Rise to investigate food web alterations caused by salp blooms. Salps are gelatinous filter feeders with exceptionally high filtration rates and large predator:prey size ratios[20,24,25]. They have an "alternation-of-generations" life cycle with a chain-forming ("aggregate") sexual stage and a

solitary asexual stage, which allows explosive population growth when conditions are favorable[26]. The episodic nature of their blooms allows them to serve as a natural experiment for assessing the foodweb impacts of large filter feeders and hence the predictions of the compensatory-foodweb-alterations hypothesis. Extensive Lagrangian-framework experiments[27,28] afforded us the opportunity for extensive characterization of pelagic food-webs from phytoplankton through macrozooplankton in subtropical and subantarctic water masses with and without salp blooms[29]. During these 4–7-day experiments we utilized traditional production and grazing mea-surements (H[14]CO$_3^-$ uptake measurements of NPP[29,30], taxon-specific protistan grazing rate measurements paired with pigment and flow-cytometry analyses[31,32], and gut pigment mesozooplankton grazing rate measurements made on individual salps and size-fractionated zooplankton communities[29,33]) to quantify energy flow across multiple ecological guilds. We quantified plankton biomass-size spectra using flow cytometry and optical imaging of individual nano- and microplankton cells using a FlowCam[20,34]. The use of compound-specific isotopic analysis ([15]N) of amino acids further allowed us to determine the trophic positions, and hence relative proportions of herbivory in the diets, of size-fractionated

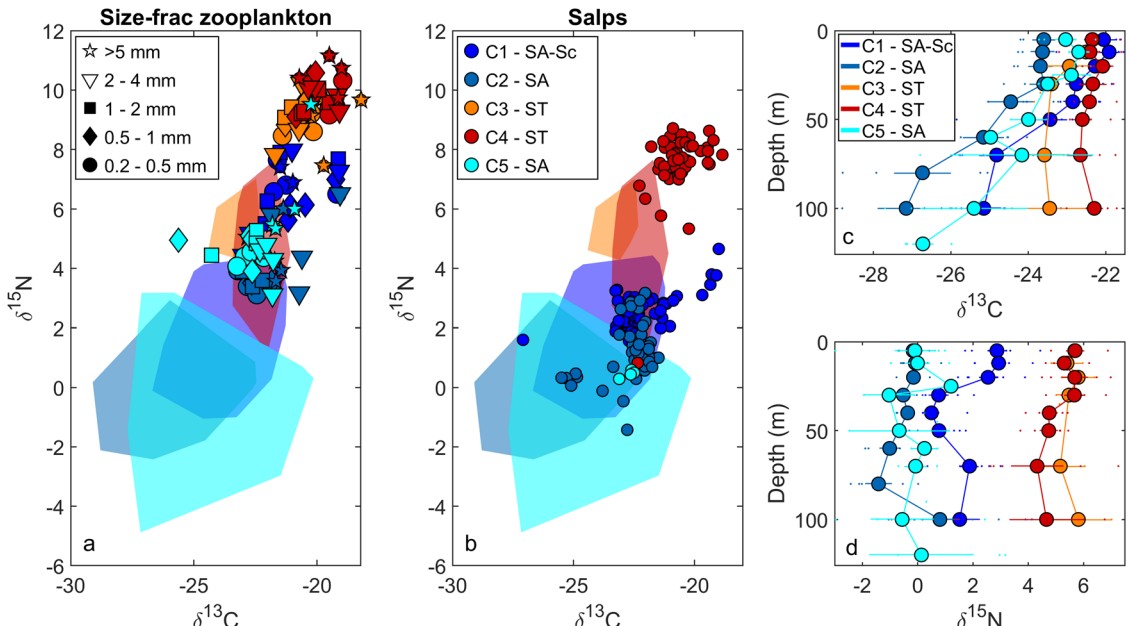

**Fig. 2 | Ecosystem bulk stable isotopes.** Ecosystem bulk stable isotopes for size-fractionated zooplankton (**a**) and salps (**b**). Color represents the Lagrangian experiment number (C1–C5). Polygons (**a**, **b**) represent the isotopic signatures of suspended particulate organic matter (POM) for the corresponding experiment.

Depth profiles are POM $\delta^{13}C$ (**c**) and $\delta^{15}N$ (**d**). Error bars are ±1 standard error of measurements made on different days of the Lagrangian experiment ($n$ = 3–6). SA Subantarctic, SA-Sc Subantarctic-Southland-Current-influenced, ST Subtropical.

(mostly crustacean) mesozooplankton (0.2 to >5 mm in size) and salps ranging in size from 10 to 132 mm. Using these detailed foodweb measurements, we tracked energy flows through different size classes in the ecosystem and find that gelatinous filter feeders substantially enhance ecosystem transfer efficiency to large organisms.

## Results

### Plankton communities of the Chatham Rise, subtropical front

We conducted five Lagrangian experiments (4–8-day duration, Cycles 1–5) near the subtropical front. Three experiments were conducted in waters with predominantly subantarctic influences (one of which was also influenced by the coastal Southland Current, Cycles 1, 2, and 5); two were conducted in subtropically influenced waters (Cycles 3 and 4; Supplementary Fig. 1). In each water type at least one experiment was conducted in a salp bloom and one "control" experiment was conducted outside of the bloom. Salp blooms were dominated by *Salpa thompsoni* (*S. thompsoni*), which ranged in size from ~6–60 mm in the sexual aggregate phase and 20–150 mm in the asexual solitary stage, although several other species were present with *Thalia democratica* contributing substantially to abundance (but not biomass) in the subtropical Lagrangian experiment and *Thetys vagina* making a small (but not negligible) contribution to biomass in subantarctic experiments. *S. thompsoni* aggregate stage biomass was 0.8 ± 0.3, 0.3 ± 0.1, and 1.0 ± 0.2 g C m$^{-2}$ and solitary stage biomass was 0.3 ± 0.1, 0.1 ± 0.0, and 0.02 ± 0.00 g C m$^{-2}$ for the Subantarctic-Southland-Current-influenced, Subantarctic, and Subtropical Lagrangian experiments, respectively.

Phytoplankton and protistan community dynamics (biomass, size spectra, species composition, NPP, and protistan grazing rates) were determined more by water mass type than the presence or absence of salps[29]. On a biomass basis, phytoplankton communities were typically dominated by nanoflagellates, except in the Southland-Current-influenced region where microplankton comprised a substantial portion of biomass. Subtropical experiments had high prymnesiophyte biomass (as evidenced by the diagnostic pigment 19-hexanoyloxyfucoxanthin), while the subantarctic communities were more variable with diatoms important in the Southland-Current-influenced region, and *Synechococcus* particularly abundant in the non-Salp Subantarctic experiment. On a carbon-weighted basis, the median

phytoplankton size was ~10 μm (equivalent spherical diameter) across the region[20].

NPP, protistan grazing, size-fractionated mesozooplankton grazing, and salp grazing measurements (previously presented in ref. 29) allow us to investigate energy flows through the planktonic food web. NPP ranged from 233 ± 44 to 747 ± 102 mg C m$^{-2}$ d$^{-1}$ across the region, and in all Lagrangian experiments protistan grazers consumed at least 69% of NPP. Non-salp metazoan grazing was substantially lower and consumed 5–21% of NPP. During Lagrangian experiments conducted within salp blooms, salp grazing rates were consistently higher than non-salp metazoans, but lower than protistan zooplankton. Salps consumed 31–50% of NPP. Across all experiments, NPP and phytoplankton growth rates were highest in the surface mixed layer.

### Stable isotopes and trophic positions of zooplankton

Bulk stable isotopes identified two important patterns (Fig. 2, Supplementary Data 1–3). First, $\delta^{15}N$ was substantially different between communities in subantarctic versus subtropical water. Seston (suspended particles including phytoplankton, phagotrophic protists, heterotrophic bacteria, and detritus), crustacean zooplankton, and salps were all enriched in $^{15}N$ in subtropical relative to subantarctic waters suggesting different source nitrogen to these ecosystems. Second, $\delta^{15}N$ of size-fractionated zooplankton samples were enriched by 4.2‰ ± 1.0‰ (mean ± st.dev.) relative to seston, while salps were enriched by 1.0‰ ± 1.4‰ relative to seston. Using a trophic discrimination factor (TDF)—the enrichment in $\delta^{15}N$ between predator and diet—of 3.4‰[35] for zooplankton implies an average trophic position based on bulk $^{15}N$ (TP$_{bulk}$) of ~2.2. However, the real TDF for zooplankton in the field is likely lower, as found by multiple other studies and supported here, as described below by compound-specific isotopic analyses to determine trophic positions.

To assess zooplankton trophic positions, we used compound-specific isotopic analysis of amino acids (CSIA-AA, Supplementary Data 4, 5). CSIA-AA determines trophic position by comparing the $\delta^{15}N$ of "trophic" amino acids that are known to enrich with subsequent trophic steps to the $\delta^{15}N$ of "source" amino acids that mostly reflect the nitrogen isotopic composition of the nitrogen source (including, e.g., upwelled nitrate and/or diazotrophy) supporting the ecosystem[36]. The CSIA-AA approach relies on

**Fig. 3 | Zooplankton and salp trophic positions.**
Trophic positions of size-fractionated zooplankton
samples (**a**) and salps (**b**). In boxplot (**a**), central red
line indicates median, box indicates one quartile
above and below median and whiskers extend to
most extreme non-outlier samples. Outliers (1.5
times the interquartile range above or below the 25th
or 75th percentile) are plotted as "+" symbols. In
(**b**), colors represent the Lagrangian experiment and
shapes represent salp species (*Salpa thompsoni*,
*Thetys vagina*, *Pegea confoedereata*, and *Soestia
zonaria*).

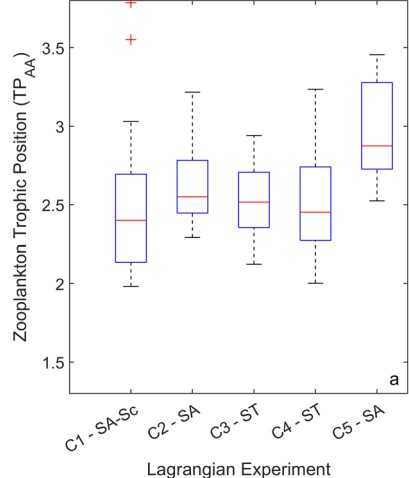
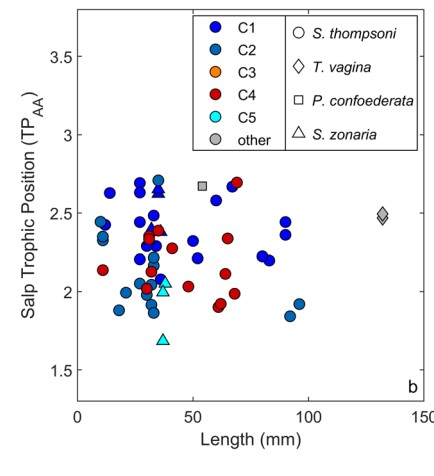

a trophic enrichment factor (TEF) which represents the relative enrichment
of trophic amino acids with respect to source amino acids compared to the
relative enrichment of these in the prey items, i.e., the difference in TDFs for
the trophic and source AAs ($TEF_{tr-sr} = TDF_{tr} - TDF_{sr}$)[37]. The TEF for
trophic and source AAs typically averages ~6‰ in aquatic ecosystems[38],
although it can vary substantially for taxa as a result of variability in
metabolic pathways, feeding modes, or prey type[39–41]. Consequently, we
compared amino acid $\delta^{15}N$ of salp tissue and salp gut contents and com-
puted a TEF specific to salps of $1.4 \pm 0.3$‰ and used this value to quantify
isotopic enrichment associated with salp trophic steps (Supplemen-
tary Fig. 2).

No significant relationship was found between salp size or species and
trophic position, which was not surprising given similarities in prey size
spectra for the different species and sizes of salps encountered on our
cruise[20,31]. However, salp trophic positions were significantly different
during Cycle 1 relative to Cycles 2 and 4 (Kolmogorov–Smirnov test,
Fig. 3b). Mean trophic positions were $2.4 \pm 0.2$, $2.0 \pm 0.3$, and $2.2 \pm 0.2$ for
Cycles 1, 2, and 4, respectively (n.b., Cycles 3 and 5 had very few salps).
Similarly, the trophic position of size-fractionated (mostly crustacean)
zooplankton showed greater variability spatially than with size class
(Fig. 3a). Cycle 5 zooplankton trophic positions were significantly different
from all other cycles (Kolmogorov–Smirnov, $p = 0.0026$). The smallest size
class of zooplankton had slightly higher trophic positions than other size
classes (TP = $2.8 \pm 0.5$ for 0.2–0.5 mm; TP = $2.6 \pm 0.3$ for all other size
classes combined). These differences were significant when comparing the
0.2–0.5 mm size class to the 1–2 and 2–5 mm size classes ($p = 0.023$ for
each). Perhaps most importantly, during Lagrangian experiments with salps
present, salp trophic positions were always lower than those of size-
fractionated zooplankton (differences were 0.20, 0.58, and 0.34 trophic steps
for Cycles 1, 2, and 4, respectively). This lower trophic position was despite
substantially larger salp sizes (10–132 mm for salps sampled for CSIA-AA)
relative to other herbivorous zooplankton (highest grazing rates were in the
0.2–1 mm sized organisms).

Glutamic acid (often used as a "trophic" amino acid) $\delta^{15}N$ is not
enriched in protistan grazers relative to their diet, while alanine $\delta^{15}N$ is
enriched. We can thus utilize differences in the enrichment of these two
amino acids to assess the mean number of protistan trophic steps within
food chains supporting metazoan zooplankton[41]. We found that the mean
number of protistan trophic steps calculated from size-fractionated meso-
zooplankton samples averaged $0.49 \pm 0.04$. When comparing to the average
trophic position of all size-fractionated zooplankton samples ($2.6 \pm 0.4$), it is
clear that carnivory on metazoans was relatively unimportant to metazoan
zooplankton diets (relative to herbivory and protistivory). This inference is
further supported by the lack of an increase in TP with zooplankton size and

implies that phagotrophic protists were an important dietary source for
metazoan zooplankton.

### Intraguild predation
Intraguild predation of salps on protistan zooplankton was confirmed
through scanning electron microscopy of salp gut contents, combined with
FlowCam (single-celled optical imaging) of water column samples[20].
Detailed taxonomic identification was only possible for >10-μm cells and
showed that ciliates (of which most species are obligate heterotrophs,
although some are kleptoplastidic mixotrophs) and dinoflagellates (which
include heterotrophic, mixotrophic, and phototrophic taxa) were over-
represented in salp guts relative to diatoms (obligate phototrophs).

Intraguild predation of salps on protistan zooplankton did not lead to a
measurable decrease in protistan herbivory rates relative to non-salp
Lagrangian experiments. This likely relates to the importance of suspension-
feeding crustaceans in consuming nano- and microzooplankton. Indeed, the
proportion of protistan zooplankton in the diets of metazoan zoo-
plankton (inferred from trophic position) was higher than in salps (although
these differences were not statistically significant): salp diets were comprised
of $18\% \pm 21\%$ (mean ± st.dev.) protistan zooplankton; mesozooplankton
consumed $30\% \pm 13\%$ protistan zooplankton. Combined with the CSIA-
AA analyses showing an average of 0.49 trophic steps within protistan
zooplankton (i.e., protistan zooplankton carnivory on other protistan
zooplankton), this highlights the complexity and pervasive intraguild pre-
dation present in marine pelagic food webs. Such common intraguild pre-
dation (across a factor of ~10,000× variability in size, linear dimension)
should be expected to prevent trophic cascades during salp blooms.

### Gelatinous filter feeders and production-size relationships
Herbivory rates, combined with trophic position data and the knowledge
that carnivory was negligible, allow us to constrain the total ingestion rates of
each zooplankton functional group. Secondary production patterns were
similar to grazing patterns, although with slightly greater production by
metazoan zooplankton, which obtained more of their nutrition from het-
erotrophs than other groups. Protistan secondary production ranged from
111 to 220 mg C m$^{-2}$ d$^{-1}$, non-salp metazoan zooplankton ranged from 7.6
to 67 mg C m$^{-2}$ d$^{-1}$, and salp secondary production (during Lagrangian
experiments in the salp bloom) ranged from 28 to 139 mg C m$^{-2}$ d$^{-1}$. These
high secondary production rates for salps (which ranged in size from ~10 to
>100 mm) were responsible for a substantial shift in production-size rela-
tionships within the salp bloom relative to non-salp-bloom conditions
(Fig. 4). Assuming that higher trophic level organisms feed at predator:prey
ratios that range from 3:1 to 300:1, we can trace this impact of salp-mediated
food web alterations to larger organisms. Within the salp bloom, the ratio of

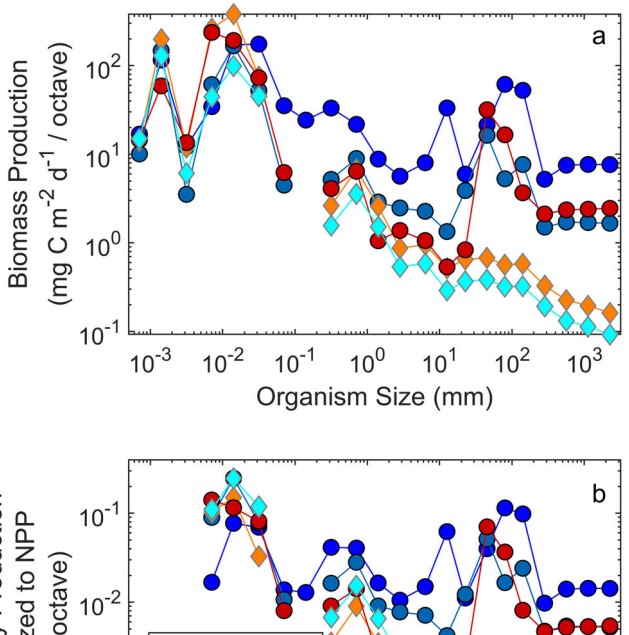

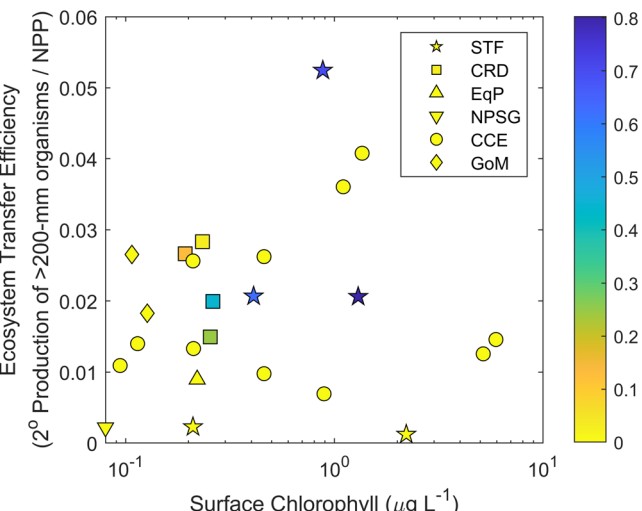

**Fig. 5 | Ecosystem transfer efficiency.** Ecosystem transfer efficiency as a function of surface chlorophyll a (x-axis) and the proportion of metazoan herbivory conducted by gelatinous filter feeders (thaliaceans = salps, doliolids, and pyrosomes), which is depicted on the color axis. Results are from this study (STF) and the Costa Rica Dome (CRD), Equatorial Pacific (EqP), North Pacific Subtropical Gyre (NPSG), California Current Ecosystem (CCE), and Gulf of Mexico (GoM). All studies had similar approaches to estimating the NPP and grazing rates of all herbivores. Trophic position was assessed via either CSIA-AA or food web models.

**Fig. 4 | Ecosystem production as a function of organism size with and without salp blooms.** Production as a function of size for different Lagrangian experiments. **a** Total biomass production (primary production + secondary production). **b** Secondary production of a size bin divided by total net primary production. To account for slightly different widths of size bins, the production of each size bin was normalized by dividing by the number of octaves (factors of 2) covered by the size bin. For phytoplankton and heterotrophic protists, we assumed that production in a size bin was proportional to biomass in that size bin (note that a gap in the 0.1–0.2 mm size bin exists for cycles in which no organisms in this size class were enumerated by FlowCam). For higher trophic level (predators of zooplankton) calculations, see the "Methods" section.

secondary production by >200 mm organisms to NPP (which we refer to as ecosystem transfer efficiency) ranged from 0.020 to 0.052, while outside the bloom it ranged from 0.0012 to 0.0023. In other words, within salp blooms, ecosystem transfer efficiency was ~2–5% of phytoplankton production, while outside the bloom it was 0.1–0.2% of phytoplankton production.

To further investigate this pattern in relation to the trophic amplification hypothesis, we compiled a larger dataset from other regions in which NPP, protistan grazing, size-fractionated mesozooplankton grazing, mesozooplankton trophic position, and gelatinous filter feeder grazing and trophic position (salps, doliolids, and pyrosomes) had been measured (Supplementary Fig. 1, Supplementary Table 1). Results showed a weak (and not statistically significant) Spearman's rank correlation between surface chlorophyll and ecosystem transfer efficiency ($\rho = 0.12$, $p = 0.59$, Fig. 5). A slightly stronger (but still not statistically significant) relationship was found between the percentage contribution of microphytoplankton to total phytoplankton biomass and ecosystem transfer efficiency ($\rho = 0.20$, $p = 0.36$). In contrast, the correlation between the proportion of metazoan herbivory attributable to gelatinous filter feeders and ecosystem transfer efficiency was substantially stronger ($\rho = 0.44$, $p = 0.03$). While the weak positive correlations associated with increased phytoplankton biomass and size do support the trophic amplification hypothesis, the stronger correlation with gelatinous filter feeders suggests that food web alterations can have a greater

impact on ecosystem transfer efficiency than changes in primary producer biomass. Furthermore, the other two instances of ecosystem transfer efficiency exceeding 3% (Fig. 5) occurred not because of especially high productivity or percentage of large phytoplankton, but rather as the result of euphausiid swarms (*Euphausia pacifica*). Euphausiids are cm-sized suspension-feeding omnivores that can feed at predator:prey size ratios exceeding 100:1. This further highlights the importance of shifts in herbivore communities (and consumers more generally) for ecosystem productivity.

## Discussion

The trophic amplification hypothesis is a robust result of end-to-end (phytoplankton to fish) ecosystem models coupled to climate simulation outputs[7]. However, these models all use fixed predator:prey size ratios for model taxa (except for one food web model that assigns the diets of each taxon individually and lacks gelatinous filter feeders). They thus lack mechanisms capable of simulating compensatory foodweb dynamics. Hence changes in food chain length are primarily mediated by changes in phytoplankton size, which is predicted by coupled climate models to be positively correlated with productivity and to decrease in much of the world ocean as a result of future increased stratification[42–44]. This oversimplifies the complex drivers that can alter food chain length and ecosystem trophic efficiency.

Multiple processes have been hypothesized to alter food chain length including resource availability[45], prevalence of intraguild predation[46], population dynamics[47], and habitat size[48,49]. Food chain length responses to potential drivers may also vary across different ecosystem types[50,51] and can impact aquatic ecosystem responses to climate change[52]. Models supporting the trophic amplification hypothesis typically suggest that food chain length will increase with decreasing productivity through the insertion of an additional protistan zooplankton trophic step when small phytoplankton replace large phytoplankton. Whether this model outcome is supported by ecological data is debatable. Ref. 53 suggested that increased energy transfer through trophic levels promotes omnivory, which in turn reduces food chain length. Similarly, an inverse relationship between food chain length (up to crustacean zooplankton) and ecosystem productivity has been found for marine ecosystems[54]. However, in semi-arid terrestrial ecosystems

predator assemblages rely disproportionately on herbivores (i.e., predators have a lower trophic level) during low productivity periods[55], a metaanalysis found no support for increasing food chain length with productivity in lakes and rivers[51], and a global pattern of sublinear-scaling of predator biomass with prey biomass suggests higher ecosystem transfer efficiency in less productive regions[56].

Intraguild predation models have been suggested as a better theoretical frameworks for investigating food chain length than linear food chains[57]. The high rates of omnivory for both size-fractionated zooplankton and salps in our study (diets averaged 31% and 23% protistan zooplankton, respectively) highlight the importance of intraguild predation in marine ecosystems. Intraguild predation can dampen ecosystem productivity responses to disturbance if the intermediate consumer is the superior competitor for the shared resource and the intraguild predator shifts to greater omnivory when productivity decreases. In the example of salps and protistan grazers, climate change mediated decreases in phytoplankton production may yield an increase in salp predation on protists (the dominant herbivores in marine systems[16]), thus acting as a negative feedback on changes in primary production. Our results support the model-derived hypothesis[58] that animal diversity can promote higher consumer biomass despite increased rates of intraguild competition. They also support the hypothesis that mobile consumers (including salps, which vertically migrate through the water column) can stabilize foodwebs[59].

In this study, we capitalized on transient dynamics using a quasi-steady state approach which is only relevant for quantifying altered trophic patterns of taxa that respond on the timescales defined by salp bloom dynamics. Thus, it is not possible with our approach to conclusively determine whether increased production of 10–100 mm salps actually translated to increased production of larger consumers. Nevertheless, our results highlight the importance of whole ecosystem studies, while elucidating variability in TDFs between taxa (e.g., salps, protists) that complicate attempts to quantify food chain length by only conducting CSIA-AA measurements on top predators without knowledge of the likely organisms comprising the food chain. Future studies will need to address these issues, while accounting for the vastly different response timescales between primary producers, intermediate consumers, and top predators in marine pelagic ecosystems. Combined use of isotopic, dietary, and size spectra data analyzed using non-steady-state frameworks may enable fruitful advances[60,61]. Whole ecosystem climate change predictions will also require consideration of variability in the metabolism, feeding ecology, phenology, reproduction, and early life stage survival of forage fish and top predators, in addition to plankton[62,63].

While our results demonstrate striking shifts in food chain length and ecosystem trophic efficiency that can occur as a result of changes in herbivore communities from cruise feeders to true filter feeders, it remains unclear whether such a shift will result from climate change. Model results suggest a future increase in gelatinous filter feeders[23,64] and gelatinous taxa more generally have been predicted to increase in abundance due to anthropogenic impacts[65,66]. Increasing salp abundance has been observed near Antarctica[67] and a large, sustained increase in pyrosome abundance occurred off the United States west coast following a strong marine heatwave[68]. Nevertheless, reliable gelatinous zooplankton timeseries are scarce, because such taxa are poorly collected by traditional net sampling approaches[69] and exhibit very patchy distributions.

Furthermore, we assumed that gelatinous filter feeders have similar palatability to crustacean zooplankton. Early studies often considered salps to be "trophic dead ends", although subsequent research has shown them to be common prey items[70]. Indeed, salps have been identified as important dietary components of such diverse organisms as anchovies, myctophids, bluefin tuna, and the sooty shearwater[71–74] and in our study region several commercially fished species of Oreosomatidae specialize in feeding on salps and other gelatinous taxa[75]. It is possible that the fate of salp blooms (trophic link vs. export as "jelly falls") may depend on the predictability of salp blooms in an ecosystem. When salp blooms are stochastic, predators cannot adjust to their abundances and blooms terminate with massive export of carbon into the deep ocean[76]. However, where salp blooms are common and

repeatable occurrences, as on the Chatham Rise, salps are important intermediaries of ocean foodwebs. The timescales over which future blooms become predictable can also be important, although habitat might play a role as well. In the Chatham Rise, we find species of demersal fish that are specialized for the consumption of gelatinous tunicate prey, which likely evolved over a long time period[75]. While the demersal lifestyle might be better suited for fish with a "belly full of jelly"[77], schools of epipelagic fish also consume salps and doliolids[62,72], and some regions recently invaded by pelagic tunicates have seen a shift in fish diets[62]. Thus, in a future ocean with potentially greater salp abundance, the degree to which we might expect salp roles to shift even further toward a paradigm of increased trophic transfer efficiency will depend on the rate at which consumers can shift to capitalize on a gelatinous diet. Equally important, subsequent studies will need to assess not only whether gelatinous taxa are consumed by planktivores, but also how a gelatinous diet (and potentially different prey stoichiometry) affects consumer growth efficiency[15,70,78].

Our study adds to a growing body of literature suggesting complex responses of marine ecosystems to climate change. From phytoplankton to fish, physiological plasticity and compensatory changes within functional groups may offset predicted changes driven by warming and altered nutrient supply[63,79,80]. These results suggest that, rather than leading to an overall decline in fisheries production, climate change may result in a reshuffling of marine food webs with different winners and losers at each trophic level. Accurate simulation of marine ecosystems will require the inclusion of compensatory-foodweb-dynamics mechanisms in coupled climate models. Inclusion of gelatinous taxa in global circulation models has begun[81,82], although current models underestimate the temporal periodicity of thaliacean blooms and focus only on the lower foodweb. Validation of such models will require adequate time-series analyses that can resolve changes in gelatinous zooplankton abundances[83] and whole ecosystem studies that quantify shifts in energy transfer in response to modified herbivore communities.

## Methods
### Sample collection

Samples were collected during the SalpPOOP cruise[29] on the R/V Tangaroa during October and November 2018. We used a Lagrangian experimental approach to follow water parcels for a period of 4–8 days, while repeatedly sampling plankton communities. The Lagrangian approach was achieved using two drifting arrays equipped with satellite-enabled surface floats and $3 \times 1$-m holey sock drogues at a depth of 15 m[27].

Phytoplankton and microbial communities were sampled daily (~2:00 a.m. local time) with a CTD-Niskin rosette. Cyanobacteria and picoeukaryote abundances were determined by flow cytometry (Accuri C6 flow cytometer used at sea for enumeration of <4-μm eukaryotic phytoplankton and *Synechococcus* from live samples; Beckman Coulter CytoFLEX S flow cytometer used on land for enumeration of *Prochlorococcus* from preserved samples[20]. Accuri C6 phytoplankton were delineated by bit-map gating of clusters on chlorophyll (FL3-A) vs. phycoerythrin (FL2-A), with *Synechococcus* showing positive FL2-A fluorescence and other groups having negative FL2-A fluorescence (Supplementary Fig. 4a), and the remaining phytoplankton subdivided into approximate sizes using fluorescent-bead calibrated FSC-A signals (Supplementary Fig. 4b, c). *Prochlorococcus* abundance (CytoFLEX S) was from a cluster sequentially gated on phycoerythrin (negative) vs. DNA (positive) (Supplementary Fig. 4d), followed by chlorophyll (positive) vs. DNA (positive) (Supplementary Fig. 4f–h), then further refined using its scatter signals, as detailed in ref. 84. Greater than 4-μm phytoplankton were imaged and enumerated using a FlowCam model VS-IV's with 10× objective lens after concentration by gravity filtration[20]. Phytoplankton diagnostic pigments were measured by high-performance liquid chromatography (HPLC) as detailed in ref. 29. NPP was measured using the $H^{14}CO_3^-$ uptake approach daily at 6 depths spanning the euphotic zone. At each depth, triplicate samples (250 mL) plus a dark bottle to quantify non-photosynthetic $H^{14}CO_3^-$ uptake were gently filled from Niskin bottles (using acid-cleaned silicon tubing) and incubated for

24 h on one of the in situ arrays at the depth of collection[29]. Taxon-specific phytoplankton mortality due to protistan grazing was quantified daily using two-point grazing dilution experiments[85]. Incubation bottles were similarly sampled from Niskin bottles and incubated on the in situ array at 6 depths[29]. Particle stable isotopes ($\delta^{13}C$ and $\delta^{15}N$) were measured on samples (2.2 L) collected using the CTD rosette, filtered onto 25-mm pre-combusted GF/F filters, dried at 60 °C for 24 h, and stored in a desiccator for analyses on land.

Zooplankton were sampled using a combination of nets for different purposes. Bongo nets (0.7-m diameter, 200-µm mesh, double oblique tow to a depth of 200 m) were towed at least twice daily (day and night) to quantify zooplankton abundances in the epipelagic zone[29]. Salps were removed from tows and sorted to species before preservation for either grazing rate measurements (via the gut pigment method) or isotopic analyses (bulk and/or CSIA-AA). A fraction of each tow was size-fractionated through nested sieves (5, 2, 1, 0.5, and 0.2 mm) and immediately frozen for gut pigment analyses of the mesozooplankton community (mostly crustaceans). A second fraction was similarly size-fractionated and frozen at −80 °C for biomass and isotopic analyses. A Multiple Opening and Closing Nets and Environmental Sampling System (MOCNESS) was used to collect depth-stratified salp abundance data and confirmed that salps were primarily present within the euphotic zone. A "salp net" (equipped with a 20-L non-filtering cod end) was used to collect live salp specimens for incubation experiments which confirmed that salps were primarily feeding on nano-phytoplankton, with a commensurately high predator:prey size ratio[31]. For additional methodological details about net tows and grazing rate calculations see ref. 29.

## Analytical methods

GF/F filters for particle analyses were dried at 60 °C for 24 h and sent to the UC Davis Stable Isotope Facility for bulk and CSIA analysis. Size-fractionated zooplankton were thawed, dried at 60 °C for at least 24 h, and weighed for biomass estimates. The dry zooplankton were then homogenized using a mortar and pestle, transferred to a pre-combusted glass vial, and sent to the UC Davis Stable Isotope Facility for bulk and CSIA-AA measurements. For salps, we excised the guts prior to drying and homogenization. Salps were removed from the −80 °C freezer and thawed until the sample was sufficiently malleable, but still frozen, to have the gut contents cleanly excised using a scalpel and forceps. Salp bodies and tissues were subsequently dried separately in a drying oven at 60 °C for ~48 h (because of their high moisture), homogenized with a mortar and pestle, transferred to a pre-combusted glass vial and sent to the UC Davis Stable Isotope Facility. Detailed methods used by this facility can be found on their website (https://stableisotopefacility.ucdavis.edu/) and we describe briefly: bulk stable isotopes were analyzed using an elemental analyzer interfaced to a continuous flow isotope ratio mass spectrometer (EA-IRMS), using reference materials interspersed among the samples (with a long term standard deviation of 0.2‰ for $^{13}C$ and 0.3‰ for $^{15}N$), and expressed relative to international standards Vienna Pee Dee Belemnite (VPDB) and air for carbon and nitrogen, respectively. CSIA-AA was determined using GC-combustion isotope ratio mass spectrometry (GC-C-IRMS) following prior protein hydrolysis and derivatization. Samples were analyzed in duplicate (two injections) with further injections included if measurements fell outside of the expected error (1‰). Initial AA isotopic values were adjusted using the internal standard (norleucine) co-injected with each sample, and further adjusted against values from a suite of AAs of known $\delta^{15}N$ injected every five samples.

## Trophic position calculations

Trophic position estimations based on nitrogen stable isotopes rely on the increase in $\delta^{15}N$ between a consumer and its prey. This increase is called the TDF, which in bulk tissues is typically ~3.4‰ per trophic level although this varies substantially[35]. This approach requires knowing the $\delta^{15}N$ of both the consumer and the base of the foodweb, which is difficult to obtain in dynamic pelagic systems where microscopic prey has high variability and fast turnover times, such that slower growing consumer tissues often lag in

reflecting their trophic position with respect to their prey. The use of $\delta^{15}N$ of amino acids circumvents these limitations by providing the value of both the consumer and the base of the foodweb through the differential enrichment of "trophic" amino acids (which enrich with each trophic step) and "source" amino acids which enrich minimally with each trophic step. The trophic position can be calculated as:

$$TP_{AA} = \frac{\delta^{15}N_{Tr} - \delta^{15}N_{Src} - \beta}{TEF_{eco}} + 1 \quad (1)$$

where $\delta^{15}N_{Tr}$ and $\delta^{15}N_{Src}$ are the average $\delta^{15}N$ of trophic and source amino acids, respectively, $\beta$ is the typical enrichment of trophic amino acids (relative to source amino acids) in primary producers, and $TEF_{eco}$ is the TEF, which is the difference in TDFs of trophic and source amino acids. Simply put, the TEF reflects the average increase in trophic amino acids relative to the source amino acids of consumers, relative to the primary producers in the ecosystem. Multiple combinations of source and trophic amino acids have been used in the literature. Here, we used glycine and phenylalanine as source amino acids and alanine, glutamic acid, and leucine as trophic amino acids, with an ecosystem TEF = 5.7, and $\beta$ = 3.6[38].

To estimate TDFs for salp amino acids (and the difference between the trophic and source TDFs to estimate $TEF_{salp}$), we dissected salps and removed their guts, allowing us to separately analyze body tissue and gut content for bulk $\delta^{15}N$ and/or CSIA-AA. Since gut isotopic values are likely representative of the isotopic values of prey, we used the difference in amino acid $\delta^{15}N$ between bodies and guts as $TEF_{salp}$ (Supplementary Fig. 2). These values were quite variable, likely because $\delta^{15}N$ of a salp body is an integrated measure over the lifetime of a salp (likely ~1 month in our study region[86]), while $\delta^{15}N$ of a salp gut integrates over the ~1 day gut turnover time of salps in our study region[29]. This supposition is supported by slopes of <1 in the relationships between salp body $\delta^{15}N$ as a function of salp gut $\delta^{15}N$ (Supplementary Fig. 3). In other words, salps that had recently fed on relatively $^{15}N$-enriched prey showed a lower apparent TDF than salps feeding on relatively $^{15}N$-depleted prey. An additional caveat to this approach is that it assumes all particles are equally likely to be assimilated which is likely not true. Salps have assimilation efficiencies ~0.7[87], which means about 1/3 of these contents will be egested. If the portion of particles egested is high in $\delta^{15}N$, which would be expected from more refractory particles that are less nutritious, then the real TDF would be higher than estimated here. However, the general overlap of salp tissue with both water column and gut stable isotopes support generally low TDFs. We thus believe that, despite the spread in the values (Supplementary Fig. 2) and the caveats noted, mean TDFs calculated using this approach are useful in estimating the lifetime trophic position of an individual salp. Our calculated TDFs for both bulk and trophic amino acids (Supplementary Table 2) were notably lower than typically assumed for aquatic organisms[36,88]. They agree, however, with previous results showing low bulk $\delta^{15}N$ enrichment for salps (1.0 ± 0.3) and low bulk $\delta^{15}N$ values for pelagic tunicates generally[89–91]. For the specific combination of trophic and source amino acids listed above, we calculated a TEF for salps ($TEF_{salp}$) of 1.0 ± 0.4‰.

Because the TEF for salps was distinctly different from TEFs that are common for most aquatic organisms, we introduce an equation specific for calculating the trophic position of salps (or organisms known to be relying on a salp food chain):

$$TP_{AA,Salp} = \frac{\delta^{15}N_{Tr} - \delta^{15}N_{Src} - \beta - TEF_{salp}}{TEF_{eco}} + 2 \quad (2)$$

This equation is equivalent to Eq. 1, but assumes that a trophic step including salps only leads to an increase in the difference between trophic and source amino acids of $TEF_{salp}$.

We also quantified the mean number of trophic steps within protistan zooplankton using the observation that alanine is enriched by protistan trophic steps, while glutamic acid is not[41]. The mean number of protistan trophic steps in a food chain including a specific consumer can then be

calculated as the difference between $TP_{ala-phe}$ and $TP_{glx-phe}$ where $TP_{ala-phe}$ and $TP_{glx-phe}$ are calculated from Eq. 1, using $TDF_{ala} = 4.5$, $TDF_{glx} = 6.1$, $\beta_{ala} = 3.2$, and $\beta_{glx} = 3.4$[92]. We calculated the mean number of trophic steps for size-fractionated zooplankton samples, but did not calculate it for salps, because glutamic acid $\delta^{15}N$ was not substantially elevated in salps relative to their guts.

### Prokaryote and protist size distributions

We quantified the size distributions of phytoplankton and heterotrophic protists using a combination of flow cytometry (for <4 μm cells) and FlowCam (for >4 μm cells). Samples for cyanobacteria were analyzed using a Beckman Coulter CytoFLEX S flow cytometer. Taxa were distinguished based on their fluorescence and light-scattering characteristics and were assumed to have a cell size between 0.5 and 1 μm for *Prochlorococcus* with a biomass of 36 fg C cell[−1] and a cell size between 1 and 2 μm for *Synechococcus* with a biomass of 255 fg C cell[−1][93]. Samples for <4 μm eukaryotic phytoplankton were analyzed using an Accuri C6 Plus flow cytometer at sea. Cell diameter was estimated from forward light scatter calibrated with plastic beads. Samples for >4 μm cells were imaged using a FlowCam (model VS-IV) with 10× objective and vignettes were manually sorted into the finest possible taxonomic resolution. For many naked nanoflagellate cells (mostly in the 4–10 μm size range) taxonomic assignment was not possible and we assumed that 2/3 of these cells were phytoplankton and 1/3 were heterotrophic protists. To estimate sizes of eukaryotes we mostly used equations from ref. 94, although for a few specific minor groups we used different carbon:volume conversions as outlined in ref. 20.

### Calculating production as a function of size and trophic position

To calculate the trophic position and secondary production of protistan grazers, we assumed that excess production of herbivorous protists that is not consumed by metazoan zooplankton must be dissipated through multiple protistan trophic steps. Thus we can write that:

$$Protistan\ Grazing = Dis_{Pro} + Pred_{Zoo,Pro} + Pred_{Salp,Pro} \quad (3)$$

where $Pred_{Zoo,Pro}$ and $Pred_{Salp,Pro}$ are the predation rates of zooplankton and salps, respectively, on heterotrophic protists. $Dis_{Pro}$ is the energy dissipation of the heterotrophic protist community (i.e., sum of respiration, excretion, and defecation). Another equivalent way of defining DisPro is as the product of the total ingestion by heterotrophic protists multiplied by one minus the gross growth efficiency of heterotrophic protists. If we assume that the protistan community is supported by phytoplankton production, we can write that:

$$Dis_{Pro} = \left(1 - GGE_{Pro}\right) \times Protistan\ Grazing \times \sum_{2}^{TP_{Pro}} GGE_{Pro}^{TP_{Pro}-2} \quad (4)$$

Where $GGE_{Pro}$ is the gross growth efficiency of protistan grazers (assumed to be 0.3[95]), $TP_{Pro}$ is the trophic position of heterotrophic protists and ProtistanGrazing is our measured protistan grazing rate on phytoplankton. In this equation, the product of ProtistanGrazing times the summation term is equal to the total consumption of heterotrophic protists. Using equations to solve a geometric series, we can simplify to:

$$Dis_{Pro} = \left(1 - GGE_{Pro}\right) \times Protistan\ Grazing \times \frac{1 - GGE_{Pro}^{TP_{Pro}-1}}{1 - GGE_{Pro}} \quad (5)$$

To solve for protistan trophic position, we further needed to know the predation rates of zooplankton and salps on heterotrophic protists. We quantified this using gut pigment and trophic position measurements of each group:

$$TP_{AA,Zoo} = TP_{Phy} \times DF_{Zoo,Phy} + TP_{Pro} \times DF_{Zoo,Pro} + 1 \quad (6)$$

where $TP_{AA,Zoo}$ is the mean trophic position of the zooplankton community, $TP_{Phy}$ is the trophic position of obligate photoautotrophs (equal to 1), $TP_{Pro}$ is the trophic position of heterotrophic protists (i.e., protistan zooplankton), and $DF_{Zoo,Phy}$ and $DF_{Zoo,Pro}$ are the dietary fractions of phytoplankton and heterotrophic protists, respectively. Because gut pigment measurements quantify grazing on all phytoplankton, we can write that (once converted to carbon units) they are equal to:

$$GutPig_{Zoo} = DF_{Zoo,Phy} \times TotalIngestion_{Zoo} \quad (7)$$

Note that we assumed that metazoan zooplankton did not comprise a meaningful proportion of other metazoan zooplankton diets, because size-fractionated zooplankton CSIA-AA results suggested that the difference between zooplankton trophic positions and 2 (i.e., an herbivore) was not statistically different from the average number of trophic steps within protists in food chains reaching these metazoan zooplankton. Similar equations to Eqs. 6 and 7 were also used for salp trophic positions. We solved the above equations recursively to quantify the average trophic position and secondary production of protistan zooplankton. If no solution existed to these equations (which occurred if total ingestion of heterotrophic protists by metazoans was greater than heterotrophic protist secondary production) we assumed that heterotrophic protists had a trophic position of 2.

Using the above information, we then quantified the dietary fractions for each zooplankton and salp size class by rearranging the equation:

$$TP_{AA,Zoo,i} = TP_{Phy} \times DF_{Zoo,Phy,i} + TP_{Pro} \times DF_{Zoo,Pro,i} + 1 \quad (8)$$

where subscript $i$ refers to individual zooplankton size classes. This allowed us to calculate the secondary production of size class $i$ as:

$$Secondary\ Production_{Zoo,i} = \frac{GutPig_{Zoo,i}}{DF_{Zoo,Phy,i}} \times GGE_{Zoo} \quad (9)$$

where $GGE_{Zoo}$ is the gross growth efficiency of zooplankton (assumed to be 0.3[95]). Size classes were chosen to be approximately octave scaled but modified slightly to match measurements and typical cutoffs used in other studies (i.e., instead of 0.2, 0.4, and 0.8 mm we used cutoffs of 0.2, 0.5, and 1.0 mm). Similar equations to Eqs. 8 and 9 were used for salps.

To estimate secondary production of higher trophic levels, we assumed that predator:prey size ratios of planktivores typically varied from 3:1 to 300:1. These values were considered to be representative of predators from carnivorous zooplankton (e.g., chaetognaths and siphonophores) to larval and adult planktivorous fish[17,96–99]. We modeled predator:prey size ratios as a uniform distribution (in log space). This thus assumes that the probability that a zooplankton will be eaten by a predator that is between 3 and 6 times its size is equal to the probability that it will be eaten by a predator that is 100 to 200 times its size. Using these assumptions, we assigned the secondary production of zooplankton to consumption by larger predators and then calculated the secondary production of these higher trophic levels using a gross growth efficiency of 0.3[95]. To calculate total biomass production or total secondary production (as shown in, e.g., Fig. 4) we summed the biomass production across trophic levels (e.g., the 32–64-mm size class includes the secondary production of salps in this size range plus the secondary production of carnivores that fed on smaller zooplankton).

Please note that in all the equations above, we neglected the trophic role played by heterotrophic bacteria. While bacteria play an important role in nutrient regeneration and can contribute to the secondary production of protists, their activities could not be constrained by the available measurements and their production is mostly dissipated within the microbial loop with little contribution to higher trophic levels.

### Statistics and reproducibility

Replicates shown in the manuscript are all true replicates collected on different days of a Lagrangian experiment and error bars in the figures show ±1 standard error of these daily replicates (typically four). Boxplot shows

median and quartiles with whiskers extending to most extreme non-outlier samples. Outliers (1.5 times the interquartile range above or below the 25th or 75th percentile) are plotted as "+" symbols.

## Reporting summary

Further information on research design is available in the Nature Portfolio Reporting Summary linked to this article.

## Data availability

Data from this study are available at the Biological and Chemical Oceanography Data Management Office: https://www.bco-dmo.org/project/754878. Summarized data are also available in Supplementary Data 1–5 which include: (1) bulk particulate organic matter stable isotopes, (2) bulk size-fractionated mesozooplankton stable isotopes, (3) bulk salp body and gut stable isotopes, (4) compound-specific size-fractionated mesozooplankton stable isotopes, and (5) compound-specific size-fractionated salp body and gut stable isotopes.

## Code availability

Model code is available as a zip file (Supplementary Dataset 6) that includes a Matlab live script, which can be used to generate all figures and summary data in the manuscript. It is also available on GitHub: https://github.com/mstukel/SalpEcosystemEfficiency and on Zenodo: https://doi.org/10.5281/zenodo.12700512.

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

## Acknowledgements
Many thanks to our collaborators in the SalpPOOP project, especially Sadie Mills, Florian Lüskow, Morgan Meyers, Karl Safi, and Natalia Yingling. This study was funded by the U.S. National Science Foundation awards OCE1756610 and 1756465 to M.R.S. and K.E.S., respectively and Grants OCE-0417616 and OCE-2224726 to the CCE LTER Program. It was further supported by the Ministry for Business, Innovation and Employment (MBIE) of New Zealand, by NIWA core programs Coast and Oceans Food Webs (COES), Ocean Flows (COOF), and by the Royal Society of New Zealand Marsden Fast-track award to M.D.

## Author contributions
M.R.S. and M.D. designed the study and wrote the manuscript. M.D., M.R.S., and C.K.F. were responsible for CSIA-AA measurements. M.R.S. was responsible for data and statistical analyses. K.E.S. and C.K.F. were responsible for phytoplankton measurements. A.G.R. and M.D. were responsible for protistan and metazoan grazing rates, respectively.

## Competing interests
The authors declare no competing interests.
