## [Peer Review File · Communications Biology]

Reviewers' comments:

Reviewer #1 (Remarks to the Author):

General comments

This is an interesting study that makes observations to inform the role played by in ecosystem trophic transfer efficiency. The value in the study lies in the empirical observations that can be used to inform ecosystem models. While these implications are important, they should not be the primary focus of the study, which should focus on the field of observation and measurement of such phenomena. The introduction does not give the reader a sufficient background about how observations and trophic level calculations are made, and what is know and unknown about them. The observations and calculations are valuable in themselves – this should be the focus of the introduction and discussion, with implications for wider ecosystem modelling and climate change studies.

Introduction - no mention of ecosystem efficiency despite this being in the title. Suggest introducing this concept a bit more. Suggest to cite some trophic transfer efficiency studies such as Ryther 1969, Stock et al. 2017 PNAS, and Eddy et al. 2021 TREE.

Methods - a brief description of the methods could be moved to main body of text after Introduction unless the journal requires them to be after the references. The remaining, technical methods could go in a supplement. The introduction is very short and doesn't give the reader a lot of context to interpret the results. Having the methods in the main body would help this a bit, alternatively the intro could be beefed up a bit. Personally, I jumped to the Methods section before reading the Results.

Spaces before units throughout.

Methods - unclear how the additional study locations in Supp. Fig. 1 were used in the study as they are not mentioned in the Methods section.

Discussion - the beginning of the discussion should be a summary of the results

Discussion should acknowledge that there are food web dynamics that drive ecosystem efficiency beyond plankton

Specific comments

Lines 30-31 - it would make sense to cite the IPCC AR6 here as a summary of the CMIP6 NPP projections

Lines 36-37 - it would make sense to cite du Pontavice et al. 2023 PLOS ONE here as well

Figure 1. This text should be in the main body of the manuscript and not the figure legend:
The trophic amplification hypothesis is based on conventional size-structured ecosystem models (a,b). Thus, shifts towards small phytoplankton in a future climate (represented by moving from panel a to b) leads to food chain elongation through insertion of additional protistan trophic levels. In contrast, the compensatory foodweb dynamic suggests that bottom-up processes driving a shift from large (panel c) to small (panel d) phytoplankton would be accompanied by a shift of metazoan communities towards filter feeders with large predator prey ratios (e.g., salps). This conceptualization of the foodweb involves high functional diversity amongst consumer trophic levels and substantial intraguild predation. These processes could stabilize ecosystem functions in response to climate change disruptions of nutrient supply.
The colour of the circles being proportional to production of a functional group needs to be explained more in the figure legend. How is this calculated?

Figure 2 - panels should be labelled A, B, C, D

Figure 3 - panels should be labelled

Lines 183-191 - how were these other regions chosen? What types of ecosystems are they?

Supp. Fig. 1. - Name the study locations from the left panel in the figure legend.

Line 540 - 2:00 am or pm?

Line 548 - space before unit (250 mL)

Line 555 - spaces before units (25 mm, 60 °C). Many additional place where this needs to be corrected below

Supp. Fig. 2 - Define 'AA' acronym in figure legend.

Methods section - use past tense throughout

Supp. Fig. 3 - acronyms in the legend should be defined in the figure legend

Reviewer #2 (Remarks to the Author):

This is an important manuscript adding new evidence to the complexity of marine food webs with implications for projection of potential climate effects. By using experimental measurements (stable isotope and production fluxes) the authors conclude that the trophic efficiency increases during salp blooms independently from changes in primary productivity mainly because of the enhanced channeling of most primary production through these gelatinous organisms. The

addition of trophic steps through protistan grazers appears as a compensatory mechanism rather than a negative effect on trophic efficiency caused by decreases in phytoplankton cell size and productivity. These conclusions support a major role of intraguild predation and protist consumption in regulating marine pelagic food webs while dismiss the prevailing view of a general amplification of trophic effects to upper trophic levels caused by the climatic-related decreases in phytoplankton size. While this study does not consider upper trophic levels, the conclusions may also be relevant for entire food webs. This work can be of use in the design of future experimental studies and models aiming to predict the response of pelagic food webs to changes in multiple factors, including productivity, diversity, and body size of planktonic organisms.

The manuscript is concise and contains all the information required to support the authors' interpretation. However, the potential implications for plankton consumers could have been addressed in a more comprehensive way (beyond the brief mention in L285-286) and the changes in food chain length backed up by additional estimates.

For instance, a recent review of trophic position estimates showed that the importance of protistan trophic steps decreases as trophic position increases in marine pelagic food webs being less relevant for trophic positions above 3.9 for fish (Viana et al., 2023).

In addition, the authors' may consider independent estimations of food chain length based on the plankton size distributions (e.g. Zhou 2006, Basedow et al. 2016) to further support their interpretations based on stable isotopes.

Minor comments:

L 21: "dominance": specify if it refers to abundance (number) or biomass

L 104 and L 113 (and L 591-653 in the Methods supplement). Why using TDF and TEF? Both are observed net isotopic fractionation factors (i.e. including all molecular and tissue fractionation steps). Both are used indistinctly in the relevant literature (e.g. TDF for amino acids in Ohkouchi et al. 2017, Bradley et al., 2015, McMahon and McCarthy 2016; but TEF in other amino acid studies: Germain et al. 2013, Nuche-Pascual et al. 2021; also TEF in bulk studies: Fry 2006, Hunt et al. 2015; but TDF in Bloomfield et al. 2011). Consider using only one acronym and subscripts (as in Ohkouchi et al. 2017) when referring to the trophic enrichment.

L 112; the sentence may be wrongly interpreted as "diazotrophy = nitrogen isotopic composition of the nutrients". Change or modify, e.g.: (including diazotrophy and nitrification)

L 143: "protistivory" may be considered a type of carnivory (grazing on protozoa). Use other term instead of carnivory in this context, e.g. grazing on metazoa

L 157-158: are these protistan diets for salps and mesozooplankton statistically significant?

L 286. Consider the upper limits for detecting protistan trophic steps in Viana et al. (2023).

Additional references

Basedow, S. L., de Silva, N. A. L., Bode, A. & van Beusekorn, J. Trophic positions of mesozooplankton across the North Atlantic: estimates derived from biovolume spectrum theories

and stable isotope analyses. *J. Plankton Res.* 38, 1364-1378 (2016).

<https://doi.org/10.1093/plankt/fbw070>

Bloomfield, A. L., Elsdon, T. S., Walther, B. D., Gier, E. J. & Gillanders, B. M. Temperature and diet affect carbon and nitrogen isotopes of fish muscle: can amino acid nitrogen isotopes explain effects? *Journal of Experimental Marine Biology and Ecology* 399, 48-59 (2011).

<https://doi.org/10.1016/j.jembe.2011.01.015>

Fry, B. *Stable isotope ecology*. (Springer Science+Business Media, LLC, New York, 2006).

Germain, L. R., Koch, P. L., Harvey, J. & McCarthy, M. D. Nitrogen isotope fractionation in amino acids from harbor seals: implications for compound-specific trophic position calculations. *Mar. Ecol. Prog. Ser.* 482, 265-277 (2013). <https://doi.org/10.3354/meps10257>

Nuche-Pascual, M. T., Ruiz-Cooley, R. I. & Herzka, S. Z. A meta-analysis of amino acid $\delta^{15}\text{N}$ trophic enrichment factors in fishes relative to nutritional and ecological drivers. *Ecosphere* 12, e03570 (2021). <https://doi.org/10.1002/ecs2.3570>

Viana, I. G., García-Seoane, R. & Bode, A. The missing trophic link: Contribution of the microbial loop to the estimation of the trophic position of pelagic consumers. *Limnol. Oceanogr.* 68, 2587-2602 (2023). <https://doi.org/10.1002/lno.12445>

Zhou, M. What determines the slope of a plankton biomass spectrum? *J. Plankton Res.* 28, 437-448 (2006). <https://doi.org/10.1093/plankt/fbi119>

We would like to thank our reviewers for their helpful comments. As you will see below, we have made many modifications to the manuscript in response to these comments.

Reviewer #1 (Remarks to the Author):

General comments

This is an interesting study that makes observations to inform the role played by in ecosystem trophic transfer efficiency. The value in the study lies in the empirical observations that can be used to inform ecosystem models. While these implications are important, they should not be the primary focus of the study, which should focus on the field of observation and measurement of such phenomena. The introduction does not give the reader a sufficient background about how observations and trophic level calculations are made, and what is know and unknown about them. The observations and calculations are valuable in themselves – this should be the focus of the introduction and discussion, with implications for wider ecosystem modelling and climate change studies.

We disagree that the observations and measurements – and methodology used to obtain them – are more interesting to potential readers of our manuscript than the implications that can be derived from these observations and measurements. We prefer the current structure of our manuscript, in which the ecological theory we are investigating provides the rationale for our study, while a very detailed methodological section outlines how we investigated these ecological theories. We thus prefer not to make methodology and calculations the centerpiece of the introduction section. However, we have added substantial methodological information to the end of the introduction section.

Introduction - no mention of ecosystem efficiency despite this being in the title. Suggest introducing this concept a bit more. Suggest to cite some trophic transfer efficiency studies such as Ryther 1969, Stock et al. 2017 PNAS, and Eddy et al. 2021 TREE.

Thank you for pointing this out. We have added a paragraph to the introduction explaining ecosystem efficiency.

Methods - a brief description of the methods could be moved to main body of text after

Introduction unless the journal requires them to be after the references. The remaining, technical methods could go in a supplement. The introduction is very short and doesn't give the reader a lot of context to interpret the results. Having the methods in the main body would help this a bit, alternatively the intro could be beefed up a bit. Personally, I jumped to the Methods section before reading the Results.

The Communications Biology format places methods sections at the end of the manuscript (rather than after the introduction). We agree that this can make it more difficult to understand the results (although readers can, of course, skip to the methods section as the reviewer did). Hence, we added additional details outlining our broad methodological approach to the final paragraph of the introduction section.

Spaces before units throughout.

We have inserted spaces before units.

Methods - unclear how the additional study locations in Supp. Fig. 1 were used in the study as they are not mentioned in the Methods section.

These were used in the investigation of global-scale drivers of ecosystem transfer efficiency (last paragraph of results and Fig. 5). We now note this in the legend of Supp. Fig. 1 to avoid confusion.

Discussion - the beginning of the discussion should be a summary of the results

We do not believe that the first paragraph of the discussion needs to summarize the results preceding it. Given the word limits of Communications Biology, we believe that adding text to address some of the reviewers' other concerns was more valuable than adding text to summarize the results at the beginning of the discussion.

Discussion should acknowledge that there are food web dynamics that drive ecosystem efficiency beyond plankton

This is an important point and we now state it explicitly:

“Whole ecosystem climate change predictions will require consideration of variability in the metabolism, feeding ecology, phenology, reproduction, and early life stage survival of forage fish and top predators, in addition to plankton^{58,59}.”

We also now note later in the discussion:

“Equally important, subsequent studies will need to assess not only whether gelatinous taxa are consumed by planktivores, but how a gelatinous diet (and potentially different prey stoichiometry) affects consumer growth efficiency^{15,70,78}.”

Specific comments

Lines 30-31 - it would make sense to cite the IPCC AR6 here as a summary of the CMIP6 NPP projections

We now cite a synthesis analysis of the CMIP6 studies that formed the basis for much of the relevant IPCC AR6 conclusions.

Lines 36-37 - it would make sense to cite du Pontavice et al. 2023 PLOS ONE here as well

We assume that the reviewer is actually referring to Guibourd de Luzinai et al. (2023) “Trophic amplification: A model intercomparison of climate driven changes in marine food webs.” for which Pontavice is the second author. We now cite this manuscript.

Figure 1. This text should be in the main body of the manuscript and not the figure legend: The trophic amplification hypothesis is based on conventional size-structured ecosystem models (a,b). Thus, shifts towards small phytoplankton in a future climate (represented by moving from panel a to b) leads to food chain elongation through insertion of additional protistan trophic levels. In contrast, the compensatory foodweb dynamic suggests that bottom-up processes driving a shift from large (panel c) to small (panel d) phytoplankton would be accompanied by a shift of metazoan communities towards filter feeders with large predator prey ratios (e.g., salps). This conceptualization of the foodweb involves high functional diversity amongst consumer trophic levels and substantial intraguild predation. These processes could stabilize ecosystem functions in response to climate change disruptions of nutrient supply.

Most of this is already in the first three paragraphs of the main text. For instance:

“Modeling studies have suggested a pattern of “trophic amplification”, in which declines in higher trophic levels (e.g., fish) are greater than declines in primary production⁵⁻⁸. This trophic amplification results from altered plankton size structure and commensurately longer food chains, among other processes^{7,9}.”

“The impacts of decreased net primary production (NPP) and a shift towards smaller phytoplankton can be offset, however, by foodweb alterations (Fig. 1).”

“This suggests the possibility for compensatory food-web changes in which a shift to smaller phytoplankton taxa would be accompanied by greater dominance of herbivores with a higher predator:prey size ratio. Such a shift could offset changes predicted by trophic amplification theory and stabilize food webs in response to altered nutrient supply.”

Because this concept is so central to the focus of our manuscript, we believe that it is useful to re-phrase it in the legend of our conceptual figure.

The colour of the circles being proportional to production of a functional group needs to be explained more in the figure legend. How is this calculated?

This is a conceptual diagram (which we now make more clear by defining it as a “conceptual food web diagram”). These estimates of production were made based on our own assumptions about how consumer abundances might shift in response to variability in prey taxa, coupled with knowledge of gross growth efficiencies. This is intended as a conceptual diagram, however, not a quantitative assessment (hence production colorbar is given as “low”, “medium”, and “high” rather than actual values). For this reason, it would only be distracting to go into the assumptions that we made in determining the colors.

Figure 2 - panels should be labelled A, B, C, D

Per Communications Biology guidelines, they should be labeled with lower case letters, as we have done.

Figure 3 - panels should be labelled

Labels are in the bottom right corner of each panel.

Lines 183-191 - how were these other regions chosen? What types of ecosystems are they?

Regions were chosen based on the availability of co-located measurements (in time and space) of:

- Surface chlorophyll
- Net primary production
- Protistan grazing
- Size-fractionated mesozooplankton grazing
- Mesozooplankton trophic position
- Gelatinous filter feeder grazing rates
- Gelatinous filter feeder trophic positions

These were the measurements necessary for simultaneously quantifying ecosystem transfer efficiency and the relative proportion of metazoan herbivory conducted by gelatinous filter feeders.

Ecosystem types include:

- Oligotrophic gyre (North Pacific Subtropical Gyre)
- Eastern boundary current upwelling system (California Current Ecosystem)
- Open ocean upwelling system (Costa Rica Dome)
- Open ocean upwelling system (Equatorial Pacific)
- Oligotrophic marginal sea (Gulf of Mexico)

We note that the characteristics of these ecosystems can be found in Table 6.

Supp. Fig. 1. - Name the study locations from the left panel in the figure legend.

We have added labels on the figure and names in the legend.

Line 540 - 2:00 am or pm?

am

Line 548 - space before unit (250 mL)

When a value and unit are used as a combined adjective, it is appropriate to use a hyphen rather than a space.

Line 555 - spaces before units (25 mm, 60 °C). Many additional place where this needs to be corrected below

When a value and unit are used as a combined adjective, it is appropriate to use a hyphen rather than a space.

Supp. Fig. 2 - Define 'AA' acronym in figure legend.

AA now defined as amino acids.

Methods section - use past tense throughout

Modified as suggested.

Supp. Fig. 3 - acronyms in the legend should be defined in the figure legend

Defined.

Reviewer #2 (Remarks to the Author):

This is an important manuscript adding new evidence to the complexity of marine food webs with implications for projection of potential climate effects. By using experimental measurements (stable isotope and production fluxes) the authors conclude that the trophic efficiency increases during salp blooms independently from changes in primary productivity mainly because of the enhanced channeling of most primary production through these gelatinous organisms. The addition of trophic steps through protistan grazers appears as a compensatory mechanism rather than a negative effect on trophic efficiency caused by decreases in phytoplankton cell size and productivity. These conclusions support a major role of intraguild predation and protist consumption in regulating marine pelagic food webs while dismiss the prevailing view of a general amplification of trophic effects to upper trophic levels caused by the climatic-related decreases in phytoplankton size. While this study does not consider upper trophic levels, the conclusions may also be relevant for entire food webs. This work can be of use in the design of future experimental studies and models aiming to predict the response of pelagic food webs to changes in multiple factors, including productivity, diversity, and body size of planktonic organisms.

We thank the reviewer for this positive assessment of our study.

The manuscript is concise and contains all the information required to support the authors' interpretation. However, the potential implications for plankton consumers could have been addressed in a more comprehensive way (beyond the brief mention in L285-286) and the changes in food chain length backed up by additional estimates.

We note that the original manuscript did address implications for plankton consumers in more detail two paragraphs later (in the paragraph that starts off by discuss the palatability of gelatinous zooplankton). However, we agree that this warrants greater attention and hence have expanded the discussion in both of these paragraphs (see track changes version).

For instance, a recent review of trophic position estimates showed that the importance of protistan trophic steps decreases as trophic position increases in marine pelagic food webs being less relevant for trophic positions above 3.9 for fish (Viana et al., 2023).

Thank you for pointing out this study. We now cite it at line 58, where the conclusion that the importance of microbial trophic steps declining at higher trophic levels gives additional justification for the idea of compensatory food webs.

In addition, the authors' may consider independent estimations of food chain length based on the plankton size distributions (e.g. Zhou 2006, Basedow et al. 2016) to further support their interpretations based on stable isotopes.

We do not believe that it is useful to apply the biomass-size-spectrum trophic position (TP_{BSS}) calculations to our data, because the assumptions of this approach are violated in two important ways:

- **TP_{BSS} calculates a constant trophic position for all organisms of a given size. Since our goal is to separately assess the trophic positions of salps, relative to metazoan zooplankton, this approach is not appropriate.**
- **TP_{BSS} fundamentally assumes a positive correlation between size and trophic position. In the case of salps and other metazoan zooplankton, the isotopic analyses suggest that this is not the case. Rather, large salps had equal to or lower trophic positions than substantially smaller metazoan zooplankton.**

Additionally, we would only be able to confidently apply this approach up to large metazoan zooplankton (i.e., the >5-mm size class in our size-fractionated zooplankton) because the TP_{BSS} approach assumes that the entire biomass within a size class is quantified; while our net tow approaches were optimized for crustacean and gelatinous zooplankton they are certain to miss most of the small nekton that are of similar size to salps. Although these arguments convince us not to include TP_{BSS} in our study, we do note that if included it would only strengthen our core argument. The slope of the size spectra is flatter (less negative) within the salp bloom, leading to lower trophic level TP_{BSS} calculations for macrozooplankton during the salp bloom and hence shorter predicted food chains.

Minor comments:

L 21: "dominance": specify if it refers to abundance (number) or biomass

We now specify that this is biomass dominance.

L 104 and L 113 (and L 591-653 in the Methods supplement). Why using TDF and TEF? Both are observed net isotopic fractionation factors (i.e. including all molecular and tissue fractionation steps). Both are used indistinctly in the relevant literature (e.g. TDF for amino acids in Ohkouchi et al. 2017, Bradley et al., 2015, McMahon and McCarthy 2016; but TEF in other amino acid studies: Germain et al. 2013, Nuche-Pascual et al. 2021; also TEF in bulk studies: Fry 2006, Hunt et al. 2015; but TDF in Bloomfield et al. 2011). Consider using only one acronym and subscripts (as in Ohkouchi et al. 2017) when referring to the trophic enrichment.

We agree that these terms are often confused in the literature (which is why we are careful to define them internally). However, because these are subtly (but importantly) different concepts, we believe it is most appropriate to utilize these separate terms as is commonly done in the literature.

L 112; the sentence may be wrongly interpreted as "diazotrophy = nitrogen isotopic composition of the nutrients". Change or modify, e.g.: (including diazotrophy and nitrification)

We have rephrased as: " $\delta^{15}\text{N}$ of "source" amino acids that mostly reflect the nitrogen isotopic composition of the nitrogen source (including, e.g., upwelled nitrate and/or diazotrophy) supporting the ecosystem."

L 143: "protistivory" may be considered a type of carnivory (grazing on protozoa). Use other term instead of carnivory in this context, e.g. grazing on metazoan

We rephrased as "carnivory on metazoan" to be more clear.

L 157-158: are these protistan diets for salps and mesozooplankton statistically significant?

We now state that these differences are not statistically different.

L 286. Consider the upper limits for detecting protistan trophic steps in Viana et al. (2023).

The statement here does not refer to the intrinsic limitations of the CSIA-AA approach (as applied here and in Viana et al.), but rather to the study approach of capitalizing on transient salp blooms to study altered food webs. Regardless of the sensitivity of the CSIA-AA methodology, such a transient approach cannot be accurately used to study the responses of taxa (e.g., deepwater oreos that prey upon salps in the region) whose biomass and production varies on timescales that are substantially longer than the transient bloom. Nevertheless, we thank the reviewer for pointing us to the Viana et al. manuscript, which is certainly relevant to our study.

Additional references

Basedow, S. L., de Silva, N. A. L., Bode, A. & van Beusekorn, J. Trophic positions of mesozooplankton across the North Atlantic: estimates derived from biovolume spectrum theories and stable isotope analyses. *J. Plankton Res.* 38, 1364-1378

(2016). <https://doi.org/10.1093/plankt/fbw070>

Bloomfield, A. L., Elsdon, T. S., Walther, B. D., Gier, E. J. & Gillanders, B. M. Temperature and diet affect carbon and nitrogen isotopes of fish muscle: can amino acid nitrogen isotopes explain effects? *Journal of Experimental Marine Biology and Ecology* 399, 48-59

(2011). <https://doi.org/10.1016/j.jembe.2011.01.015>

Fry, B. *Stable isotope ecology*. (Springer Science+Business Media, LLC, New York, 2006).

Germain, L. R., Koch, P. L., Harvey, J. & McCarthy, M. D. Nitrogen isotope fractionation in amino acids from harbor seals: implications for compound-specific trophic position calculations. *Mar. Ecol. Prog. Ser.* 482, 265-277 (2013). <https://doi.org/10.3354/meps10257>

Nuche-Pascual, M. T., Ruiz-Cooley, R. I. & Herzka, S. Z. A meta-analysis of amino acid $\delta^{15}\text{N}$ trophic enrichment factors in fishes relative to nutritional and ecological drivers. *Ecosphere* 12, e03570 (2021). <https://doi.org/10.1002/ecs2.3570>

Viana, I. G., García-Seoane, R. & Bode, A. The missing trophic link: Contribution of the microbial loop to the estimation of the trophic position of pelagic consumers. *Limnol. Oceanogr.* 68, 2587-2602 (2023). <https://doi.org/10.1002/lno.12445>

Zhou, M. What determines the slope of a plankton biomass spectrum? *J. Plankton Res.* 28, 437-448 (2006). <https://doi.org/10.1093/plankt/fbi119>

REVIEWERS' COMMENTS:

Reviewer #1 (Remarks to the Author):

I thank the authors for providing additional background information about ecosystem and trophic transfer efficiency and a brief summary of the methods in the main body of the text. I think that this provides additional context that strengthens the contribution. Overall, this is a nice contribution to the field that can be used to inform processes represented in ecosystem modelling and climate change projections.

Reviewer #2 (Remarks to the Author):

The authors have clarified all the issues raised by the reviewers. The revised manuscript can be now recommended for publication.